# Application of Micro-Arc Discharges during Anodization of Tantalum for Synthesis of Photocatalytic Active Ta_2_O_5_ Coatings

**DOI:** 10.3390/mi14030701

**Published:** 2023-03-22

**Authors:** Stevan Stojadinović, Nenad Radić, Rastko Vasilić

**Affiliations:** 1Faculty of Physics, University of Belgrade, Studentski trg 12-16, 11000 Belgrade, Serbia; 2Faculty of Forestry, University of Belgrade, Kneza Višeslava 1, 11000 Belgrade, Serbia; 3IChTM-Department of Catalysis and Chemical Engineering, University of Belgrade, Njegoševa 12, 11000 Belgrade, Serbia

**Keywords:** micro-arc discharges, plasma electrolytic oxidation, micro-arc oxidation, tantalum, Ta_2_O_5_, photocatalysis, optical emission spectroscopy

## Abstract

Ta_2_O_5_ coatings were created using micro-arc discharges (MDs) during anodization on a tantalum substrate in a sodium phosphate electrolyte (10 g/L Na_3_PO_4_·10H_2_O). During the process, the size of MDs increases while the number of MDs decreases. The elements and their ionization states present in MDs were identified using optical emission spectroscopy. The hydrogen Balmer line H_β_ shape analysis revealed the presence of two types of MDs, with estimated electron number densities of around 1.1 × 10^21^ m^−3^ and 7.3 × 10^21^ m^−3^. The effect of MDs duration on surface morphology, phase and chemical composition, optical absorption, and photoluminescent, properties of Ta_2_O_5_ coatings, as well as their applications in photocatalytic degradation of methyl orange, were investigated. The created coatings were crystalline and were primarily composed of Ta_2_O_5_ orthorhombic phase. Since Ta_2_O_5_ coatings feature strong absorption in the ultraviolet light region below 320 nm, their photocatalytic activity is very high and increases with the time of the MDs process. This was associated with an increase of oxygen vacancy defects in coatings formed during the MDs, which was confirmed by photoluminescent measurements. The photocatalytic activity after 8 h of irradiation was around 69%, 74%, 80%, and 88% for Ta_2_O_5_ coatings created after 3 min, 5 min, 10 min, and 15 min, respectively.

## 1. Introduction

Tantalum pentoxide (Ta_2_O_5_) is a transition metal oxide with exceptional physical and chemical properties, including high dielectric constant and refractive index, low optical absorption coefficient, excellent photoelectric performance, and good chemical stability [1,2,3]. These properties are used in a variety of Ta_2_O_5_ applications, including memory devices, coatings on photographic lenses, electrochromic devices, biocompatible materials, photocatalysts, etc. [4,5,6,7,8].

The goal of this study was to examine the possibility of synthesizing Ta_2_O_5_ coatings on tantalum substrates using micro-arc discharges (MDs) during anodization for applications in organic pollutant degradation. The process of the formation of oxide coatings under MDs is known as micro-arc oxidation (MAO) or plasma electrolytic oxidation (PEO) [9,10,11,12]. The elevated local temperature (10^3^ K to 10^4^ K) and pressure (up to 10^2^ MPa) cause numerous processes, including light and heat emission, thermal, electrochemical, and plasma-chemical reactions to take place at the MD sites [13]. The structure, content, and morphology of the produced oxide coatings are changed as a result of these processes. The coatings produced are primarily composed of substrate oxides, but more complex compounds involving the species present in the electrolyte can also be formed [10].

The formation of oxide coatings is accomplished through several steps [14]. As a result of dielectric stability loss in a low conductivity region, a number of separated MD channels are formed in the oxide layer during the first step. This region is heated to temperatures of 10^4^ K by generated electron avalanches [15]. The anionic components of the electrolyte are drawn into the channels by the strong electric field. At the same time, the metal is melted out of the substrate, enters the MD channels, and oxidizes. Plasma-chemical reactions occur in the MD channels as a result of these processes. At the same time, the presence of an electric field causes the separation of oppositely charged ions in the MD channels. Electrostatic forces eject the cations from the MD channels into the electrolyte. The oxidized metal is then ejected from the MD channels into the coating surface in contact with the electrolyte, increasing the coating thickness around the MD channels. Finally, the MD channels cool, and the reaction products are deposited on their walls. This process is repeated at a number of discrete locations across the coating surface, resulting in an increase in coating thickness. The coating material formed at the MD sites consists of crystalline and amorphous phases, with constituent species derived from both the metal and the electrolyte.

Semiconductor photocatalysis is currently regarded as one of the most promising solar energy conversion technologies for organic pollution degradation. Transition metal oxides such as TiO_2_ and ZnO are the primary photocatalysts that exhibit high efficiency for photocatalytic processes [16,17,18]. Additionally, some other transition metal oxides are used in photocatalytic processes, such as WO_3_ [19], V_2_O_5_ [20], ZrO_2_ [21], Nb_2_O_5_ [22], and Ta_2_O_5_ [23]. Ta_2_O_5_ has received a lot of attention among these transition metal oxides as a highly desirable replacement for TiO_2_ and ZnO. Because of the properly positioned band gap of Ta_2_O_5_ [24], it is possible to perform the majority of photocatalytic reactions without the use of additional components and compounds. Furthermore, its conduction band is higher than TiO_2_ and ZnO, providing more favorable conditions for the smooth continuation or initiation of reduction reactions [23]. The main disadvantage of the Ta_2_O_5_ photocatalyst is its large gap (around 4 eV), which necessitates the use of strong ultraviolet light (below 310 nm) for photocatalytic reactions [23].

The majority of studies have focused on the Ta_2_O_5_ photocatalysts in powder form, which is difficult to separate after reaction and unsuitable for recycling [25,26,27,28]. This problem can be solved by focusing on Ta_2_O_5_-based photocatalysts supported on a stable substrate. Our research has shown that efficient photocatalysts on various metal substrates can be obtained using MDs during their anodization in suitable electrolytes [29,30,31]. So far, several papers have been published that investigate oxide coatings formed on tantalum substrates under MDs, primarily for biomedical applications [32,33,34,35,36,37]. In this paper, we demonstrate for the first time that the coatings formed on tantalum substrates under MDs can be used as photocatalysts for organic pollutant degradation.

## 2. Materials and Methods

Commercial tantalum foils (99.9% purity) with a thickness of 0.25 mm were used as a starting material and were cut into samples with dimensions of 10 mm × 25 mm. Before processing, the samples were ultrasonically cleaned for 5 min in acetone, dried with a warm air stream, and covered with insulating resin to provide a working area of 10 mm × 15 mm. More details about the experimental setup can be found in Ref. [38]. In short, the electrolytic cell was a 250 mL double-walled glass cell with water cooling. The temperature of the electrolyte was kept within (20 ± 1) °C. Tantalum samples were used as anodes and positioned in the center of the electrolytic cell, surrounded by a tubular stainless-steel cathode. The process was carried out in direct current mode with a current density of 150 mA/cm^2^ for varying times up to 15 min using a Consort EV261 DC power unit. The electrolyte was a water solution containing 10 g/L Na_3_PO_4_·10H_2_O.

The morphology and elemental analyses of the coatings were performed using a scanning electron microscope (SEM, JEOL 840A, Tokyo, Japan) with energy-dispersive X-ray spectroscopy (EDS, Oxford INCA, Abingdon, UK). The phase composition of the coatings was identified by X-ray diffraction (XRD, Rigaku Ultima IV, Tokyo, Japan) using a monochromatic Cu Kα radiation. The coating surface was additionally analyzed by a X-ray photoelectron spectroscopy (XPS, SPECS System, Germany) using a monochromatic Al K_α_ radiation. Binding energies were corrected relative to the C 1s signal at 285.0 eV. The optical properties of coatings were analyzed by UV-Vis diffuse reflectance spectra (DRS, Shimadzu UV-3600, Tokyo, Japan) and photoluminescence (PL, Horiba Jobin Yvon Fluorolog FL3-22, Edison, NJ, USA) using a 450 W xenon lamp as the excitation light source.

To test the photocatalytic activity (PA) of formed coatings, the photodegradation of methyl orange (MO), as the model compound for organic pollution, was examined under artificial solar radiation at 20 °C. To achieve absorption–desorption equilibrium, a solution containing 8 mg/L of MO was left in the dark for one hour in a 6.8 cm diameter open cylindrical thermostated pyrex glass reactor. MO adsorption was minimal as evidenced by the nearly constant MO concentration. The samples were then illuminated by a 300 W lamp (OSRAM ULTRA-VITALUX UV-A, Germany) that was positioned 25 cm above the solution’s top surface. The samples were positioned 5 mm above the reactor bottom on the steel wire holder. Monitoring MO decomposition at predetermined intervals after exposure to irradiation allowed for the estimation of PA. The maximum MO absorption peak at 464 nm was used to measure the MO concentration using a UV-Vis spectrophotometer (Thermo Electron Nicolet Evolution 500, UK).

## 3. Results and Discussion

Figure 1 depicts the voltage–time response during galvanostatic anodization of tantalum. The voltage increases approximately linearly with time from the start of anodization to about 300 V in a very short time resulting in a constant rate of increase of the oxide film thickness [39]. This is followed by an apparent deviation from linearity in the voltage-time curve, beginning with the so-called sparking (breakdown) voltage. Following the breakdown, the voltage continues to rise, but the voltage–time slope drops, and a large number of small size MDs appear, evenly distributed across the entire sample surface. Further anodization results in a relatively stable value of the anodization voltage.

The process of tantalum anodization can be separated into three stages based on the voltage–time response. Total anodizing current density is the sum of ionic and electron current densities [40]. During the anodic growth in stage I, the electric field strength for a certain current density remains constant, and the ionic current is two to three orders of magnitude higher than the electronic component. To keep the electric field strength constant, the voltage of anodization must increase linearly as the film thickens. In addition, during anodization, electrons are injected into the anodic oxide’s conduction band and accelerated by the electric field, resulting in avalanches via an impact ionization mechanism [40]. The breakdown occurs when the avalanche electronic current reaches a critical value [41]. Due to the independence of electron current density with anodic oxide film thickness, a relatively low voltage is required in stage II to maintain the same total current density (compared to stage I). The fraction of electron current density in total current density becomes dominant in stage III. The total current density is almost independent of the oxide film thickness at this stage, and the voltage–time slope is close to zero.

Figure 2 depicts the appearance of MDs at various stages of the anodization process. MDs were visible after about 10 s from the onset of anodization. As the duration of the process lengthens, the size of MDs increases while the number of MDs decreases. Given that MDs are generated by dielectric breakdown through weak sites in the oxide coating, the number of weak sites decreases as anodization time, i.e., coating thickness, increases. The increased size of MDs with increasing PEO time was attributed to a reduced number of MDs sites through which higher anodic current can pass [9]. The MDs gradually vanished after 15 min, rendering the process ineffective.

Figure 3a depicts a typical optical emission spectrum of MDs in the spectral range 400 nm to 850 nm. The sodium doublet spectral lines Na I at 588.99 nm and 589.59 nm were the most intense. Additionally, detected was a relatively strong hydrogen Balmer line H_α_ (656.28 nm). The continuum emission was caused by electron collision-radiative recombination [39] and bremsstrahlung radiation [42]. There were some weak lines that corresponded to neutral oxygen atoms (O I) in the range from 700 nm to 850 nm (Figure 3c) and singly ionized oxygen atoms (O II), as well as the Balmer line H_β_ (486.13 nm), in the range from 400 nm to 500 nm (Figure 3b). However, no lines were observed from the species present in the substrate, indicating that tantalum’s high melting point prevented its evaporation during MDs.

Strong sodium doublet spectral lines [43] and Balmer line H_α_ [44] are not suitable for the spectral line shape analysis. The broadened profile of the Balmer H_β_ line was used for MDs electron number density measurements. During the analysis of the H_β_ line profile (Figure 3d), it was discovered that the H_β_ line shape could only be properly fitted using two Lorentzian profiles. The upper part of the H_β_ profile has a full width at half maximum (FWHM) of 0.23 nm, while the broad lower profile has an FWHM of 0.81 nm. This is in agreement with empirical formula (2a) in [45], which yields electron number densities of ~1.1 × 10^21^ m^−3^ and ~7.3 × 10^21^ m^−3^. Two different electron number densities most likely indicate the presence of two MDs processes: MDs in relatively small holes near the surface of the oxide layer and MDs in micro-pores near the surface of the oxide layer [46].

Top view and cross-section SEM micrographs of the oxide of coatings formed at various stages of the MDs are shown in Figure 4. The presence of pores of various diameters, shapes, and molten regions distributed throughout the surface characterize the coatings formed during MDs. The pores are associated with the release of gas bubbles from MD channels, whereas molten regions are formed as a result of heating, melting, and quenching of the molten oxide in contact with the surrounding electrolyte, resulting in an increase of surface roughness during MDs [9]. Dense layers, with an average thickness of about 5.9 μm, 8.3 μm, 9.2 μm, and 11.2 μm, were formed after 3 min, 5 min, 10 min, and 15 min of MDs, respectively.

Table 1 displays the results of the integral EDS analyses of surface coatings (the relative errors are less than 5%) shown in Figure 4a. Ta, O, and P are the primary components of the coatings. Ta substrate and electrolyte interactions during MDs lead to the observed chemical composition. P was incorporated into the coatings as a result of ${\mathrm{PO}}_{4}^{3-}$ ion movement to the Ta anode and reaction with the molten oxidized metal.

It is widely accepted that Ta_2_O_5_ has two main polymorphs, which are usually referred to as high-temperature and low-temperature phases [23]. The transition usually occurs at 1360 °C and this process is reversible, which means that the high-temperature phase cannot be stabilized. As a result, the low-temperature phase is more appealing because it can exist at ambient temperatures. It usually exists in the form of an orthorhombic or hexagonal crystal structure, with the former one being more stable. The XRD patterns of formed coatings are presented in Figure 5. Orthorhombic Ta_2_O_5_ (JCPDS, No. 25-0922) was identified as the main crystalline phase in all of the coatings. This suggests that the rapid solidification of molten Ta_2_O_5_ flowing out of the MD channels in the presence of a low temperature electrolyte promotes the formation of orthorhombic Ta_2_O_5_. The XRD pattern of coating formed for 15 min also contains a few diffraction lines with low intensities, probably connected with tantalum phosphate phases.

We used XPS measurements of coating formed for 15 min to further investigate the chemical nature, composition, and oxidation state of Ta, O, and P (Figure 6). Figure 6a depicts typical survey XPS spectra, which confirm the presence of peaks originating from Ta 4f, Ta 4d, Ta 4p, O 1s, P 2s, P 2p, and C 1. Figure 6 also includes high-resolution XPS spectra of Ta 4f, P 2p, and O 1s. The high-resolution Ta 4f XPS spectrum can be fitted into two components with peaks at 26.31 eV and 28.19 eV, which correspond to Ta 4f_7/2_ and Ta 4f_5/2_ spin orbit splitting, respectively. These peaks’ binding energy positions indicate the presence of Ta species in the form of Ta^5+^ [25]. The high resolution P 2p XPS spectrum can also be fitted into two components with peaks at 133.64 eV and 134.59 eV, which correspond to the binding energies of 2p_3/2_ and 2p_1/2_ phosphorus levels in the P^5+^ valence state [47]. The high resolution O 1s XPS spectrum can be deconvoluted into two components at 530.37 eV and 531.35 eV, indicating two distinct oxide environments: oxygen bonded to phosphorous (530.37 eV) and oxygen bonded to tantalum (531.35 eV) [48].

The photocatalytic performance of Ta_2_O_5_ coatings was tested in MO degradation. Three samples were tested for each processing time, with the mean values shown in Figure 7a. The PA of samples obtained under the same conditions is highly reproducible (within 3%). *Co* represents the initial concentration of MO, whereas *C* represents the concentration after time *t*. The coating with the highest photocatalytic efficiency was formed for 15 min. Ta_2_O_5_ coatings have a much higher efficiency in MO degradation than TiO_2_ [49], MgO [50], MgAl [38], ZrO_2_ [51], and Nb_2_O_5_ [52] coatings formed under MDs on titanium, AZ31 magnesium alloy, zirconium, and niobium substrate, respectively (Figure 7b).

The photocatalytic degradation rate of Ta_2_O_5_ coatings was calculated using the first-order kinetic Langmuir Hinshelwood model (Figure 7c):(1)$$ln\left(\frac{C_{o}}{C}\right)=k_{app}t$$

The first-order kinetic constant *k_app_* and the corresponding linear correlation coefficient (*R*^2^) are shown in Table 2. As the time of the MDs process increased from 3 min to 15 min, the *k_app_* increased from 0.146 h^−1^ to 0.262 h^−1^.

The chemical and physical stability of photocatalysts during photocatalytic reactions are critical issues for practical applications. To assess the cycling stability of Ta_2_O_5_ coatings, photodegradation on MO was cycled five times with the most active photocatalyst as a representative sample (Figure 7d). The sample was rinsed with water, dried, and reused after each photocatalytic run. Photocatalytic deactivation clearly does not occur, demonstrating the dependability and effectiveness of Ta_2_O_5_ coatings formed under MDs as photocatalysts.

The main contribution of MDs time to the PA of Ta_2_O_5_ coatings could be either in extending the optical absorption range of Ta_2_O_5_ coatings or in preventing fast recombination process of photogenerated electron/hole pairs, given that MDs time has slight effect on the phase structure of Ta_2_O_5_ coatings. Ultraviolet-visible DRS was used to investigate the light absorption properties of formed Ta_2_O_5_ coatings. As shown in Figure 8a, all of the samples have a distinct absorption band in the ultraviolet range, with the light absorption edge at about 320 nm, which is associated with the transition from the O 2p orbital to the empty Ta 5d orbitals [23], while the best ultraviolet absorption capacity has the thickest coating formed for 15 min of PEO. These findings suggest that Ta_2_O_5_ coatings can be active in photocatalytic reactions when exposed to ultraviolet light. This limits its ability to absorb only 4–5% of the sunlight. Such poor sunlight absorption capability has little practical application for the Ta_2_O_5_ photocatalysis.

The significant PA of Ta_2_O_5_ is probably related to the high concentration of different types of surface vacancies and other defects in the formed coatings during MDs. Surface vacancies, which control the electron transfer between reactants and photocatalysts, are closely related to active centers of heterogeneous photocatalytic reactions [53,54]. PL spectroscopy is an effective optical technique for detecting oxygen vacancies and other defects in semiconductor materials [55]. The PL emission spectra of formed Ta_2_O_5_ coatings excited at 260 nm are shown in Figure 8b. The PL intensity rises with increasing MDs time, suggesting that thicker coatings have higher oxygen vacancies and defect concentrations [56,57]. Throughout the PL process, oxygen vacancies and defects could bind photo-induced electrons to form free or binding excitons, allowing a PL signal to occur easily, and the higher the oxygen vacancy or defect content, the stronger the PL intensity [56,57]. However, during the photocatalysis, oxygen vacancies and defects may serve as capture sites for photo-induced electrons, effectively inhibiting photo-induced electron and hole recombination, which leads to an increase in PA of Ta_2_O_5_ coatings. Furthermore, oxygen vacancies can stimulate O_2_ adsorption, and photo-induced electrons bound by oxygen vacancies interact strongly with adsorbed O_2_ [55]. This suggests that oxygen vacancies can help adsorbed O_2_ capture photo-induced electrons while also producing ·O_2_ radical groups. The radical groups are active in promoting the oxidation of organic pollutants. Thus, oxygen vacancies and defects may favor photocatalytic reactions, and the stronger the excitonic PL intensity, the greater the oxygen vacancy or defect content, and the higher the photocatalytic activity.

## 4. Conclusions

Ta_2_O_5_ coatings were created using MDs during tantalum anodization in a sodium phosphate electrolyte for varying times. MDs were characterized using real-time images and optical emission spectroscopy. SEM/EDS, XRD, XPS, DRS, and PL were used to investigate the morphology, crystal structure, chemical composition, and optical properties of Ta_2_O_5_ coatings. The photodegradation of MO in simulated sunlight was used to assess the photocatalytic potential of Ta_2_O_5_ coatings.

The following conclusions can be drawn:

With increasing anodization time, the size of MDs increases while the number of MDs decreases. The species identified by optical emission spectroscopy under MD originate only from the electrolyte and belong to sodium, hydrogen, and oxygen. In the hydrogen Balmer line H_β_ shape analysis, two types of MDs were revealed that were connected to MDs at the oxide–electrolyte interface, with electron number densities of approximately 1.1 × 10^21^ m^−3^ and 7.3 × 10^21^ m^−3^.

The morphology of created coatings is determined by the MDs time. The chemical elements of the coatings are Ta, O, and P. Ta and P have pentavalent oxidation states, according to XPS measurements. The created coatings are crystalline and primarily comprised of the orthorhombic Ta_2_O_5_ phase. Ta_2_O_5_ coatings have a broad absorption band in the ultraviolet light region below 320 nm.

The PA of Ta_2_O_5_ coatings depends on the time of creation during MDs. The best photocatalytic performance was observed for the coating processed for 15 min. Ta_2_O_5_ coatings have a much higher efficiency in MO degradation than TiO_2_, MgO, MgAl, ZrO_2_, and Nb_2_O_5_ coatings formed under MDs on titanium, AZ31 magnesium alloy, zirconium, and niobium substrate, respectively.

## Figures and Tables

**Figure 1 micromachines-14-00701-f001:**
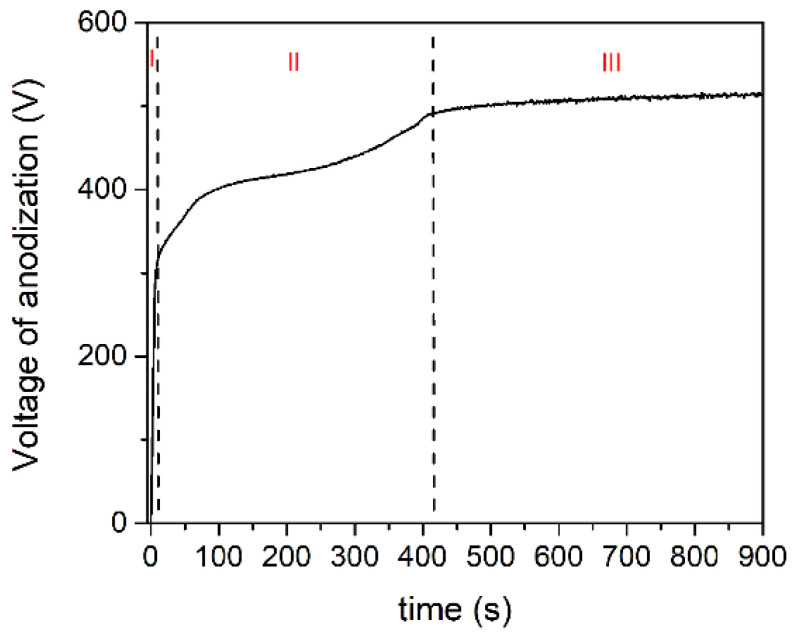
Voltage-time curve during anodization of tantalum at 150 mA/cm^2^ in Na_3_PO_4_·10H_2_O.

**Figure 2 micromachines-14-00701-f002:**
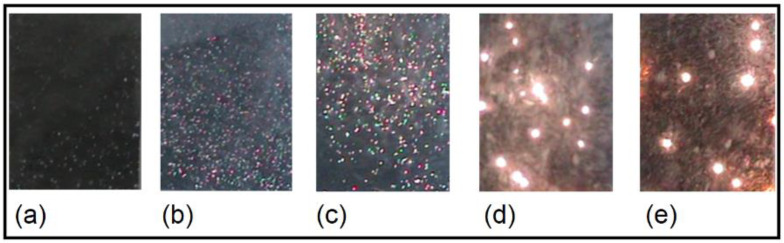
MDs appearance at various stages of tantalum anodization: (**a**) 10 s; (**b**) 150 s; (**c**) 300 s; (**d**) 600 s; (**e**) 900 s.

**Figure 3 micromachines-14-00701-f003:**
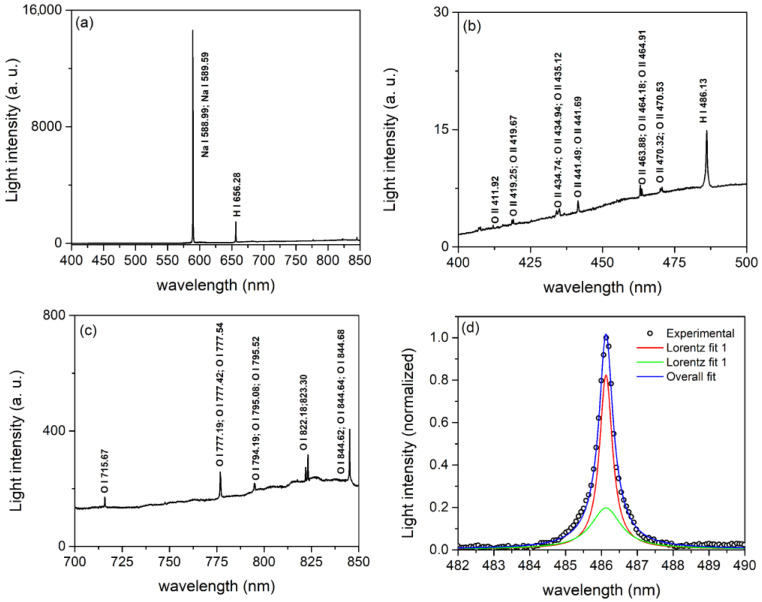
Optical emission spectra of MDs during anodization of tantalum in the range: (**a**) 400–850 nm; (**b**) 400–500 nm; (**c**) 700–850 nm; (**d**) The H_β_ line profile fitted with two Lorentzian profiles.

**Figure 4 micromachines-14-00701-f004:**
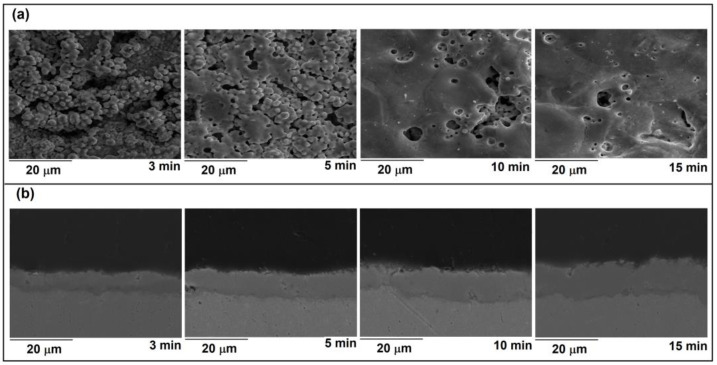
(**a**) Top view; (**b**) cross-section; micrographs of coatings formed at various stages of the MDs.

**Figure 5 micromachines-14-00701-f005:**
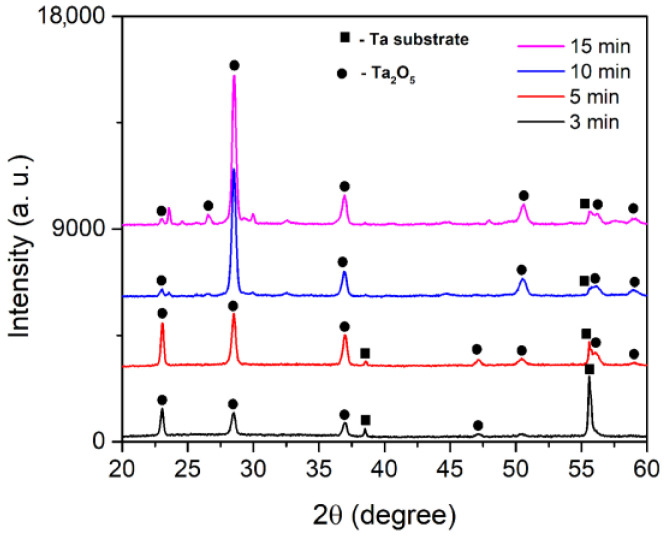
XRD patterns of coatings formed at various stages of the MDs.

**Figure 6 micromachines-14-00701-f006:**
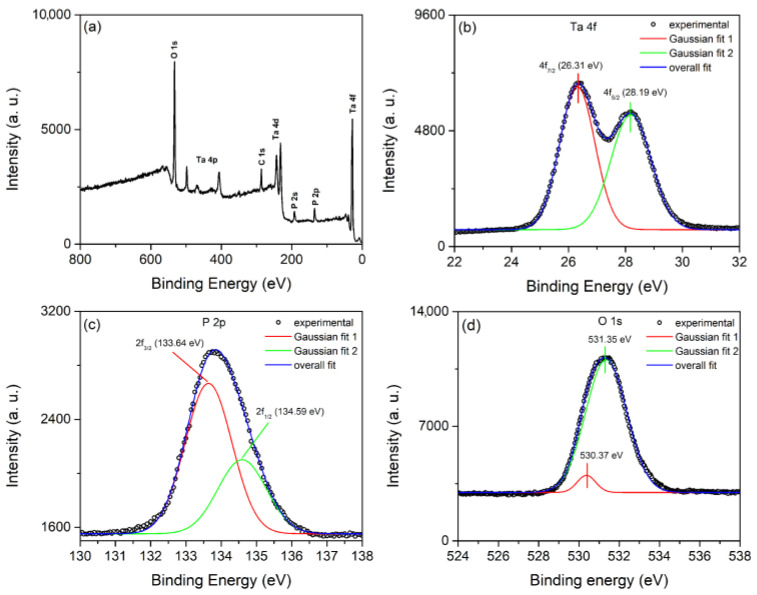
(**a**) Survey XPS spectrum of coating formed for 900 s. High resolution XPS spectra: (**b**) Ta 4f; (**c**) P 2p; (**d**) O 1s.

**Figure 7 micromachines-14-00701-f007:**
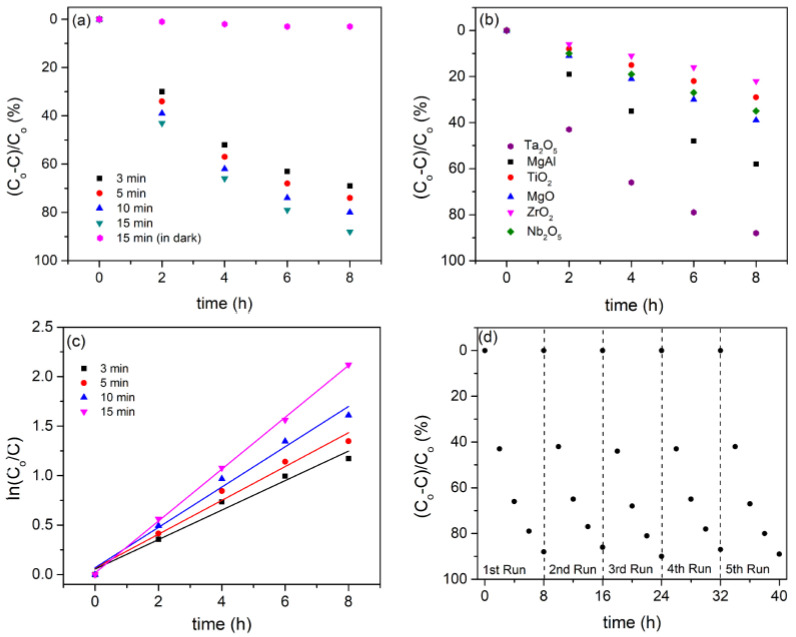
(**a**) PA of Ta_2_O_5_ coatings formed at various stages of the MDs; (**b**) PA for different coatings formed by MDs process; (**c**) First-order kinetic plots of Ta_2_O_5_ coatings formed at various stages of the MDs; (**d**) Recycling test of MO photodegradation of coating formed for 15 min.

**Figure 8 micromachines-14-00701-f008:**
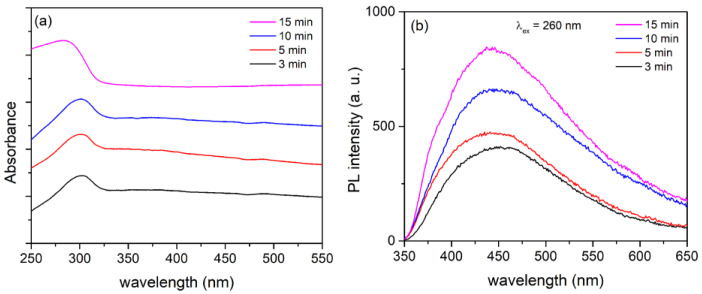
(**a**) DRS spectra; (**b**) PL emission spectra; Ta_2_O_5_ coatings formed at various stages of the MDs.

**Table 1 micromachines-14-00701-t001:** EDS elemental analysis of coatings in Figure 4a.

MDs Time (Min)	Atomic (%)		
	O	P	Ta
3	74.62	5.06	20.32
5	75.41	3.99	20.60
10	74.31	3.81	21.87
15	74.75	3.60	21.64

**Table 2 micromachines-14-00701-t002:** First-order kinetic constant *k_app_* and corresponding linear correlation coefficient *R*^2^ for Ta_2_O_5_ coatings formed at various stages of the MDs.

Time of MDs (Min)	3	5	10	15
*k_app_* (h^–1^)	0.146	0.171	0.204	0.262
*R^2^*	0.974	0.974	0.982	0.999

## Data Availability

The data presented in this study are available on request from the corresponding author.

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
