# Peer review of "Application of Micro-Arc Discharges during Anodization of Tantalum for Synthesis of Photocatalytic Active Ta2O5 Coatings"

_micromachines, 2023, doi:10.3390/mi14030701_

Round 1
Reviewer 1 Report
The authors demonstrated the fabrication of chemically stable Ta2O5 coatings on Ta substrates with micrometer thickness by using micro-arc discharges during electrochemical anodization, able to use for photocatalytic degradation of methyl orange organic compound under UV irradiation. The experiments are well-prepared and the structure of the paper is fine and has the potential to be published in the Micromachines journal. However, there are some issues that will need to be addressed before becoming suitable for publication.
1) Please include in the paper a statistical analysis, i.e. on EDX and organic pollutant decontamination section (Std. deviation or errors bars).
2) The XRD section must be improved, it looks very poor described.
3) On the XPS part a calibration information about the system must be included, the position of the adventitious C 1s position for a better understanding for the other elements states.
4) The O 1s peak from high resolution XPS spectrum could contain one more component by fitting it. This is also described by authors on the photoluminescence section where a presumption of oxygen vacancies or presence of P-O binding in the obtained metal-oxide structure is discussed.
5) From Figure 4b is not clear the morphology of obtained oxide layer. At list for 3 min and 5 min of anodization, top view (Fig. 4a) reveals a granular topography. It is expected to be distinguished also in the cross-section view. I think that authors used BSE detector in cross-section study, having much less resolution than SE detector. Can the authors clarify this issue?
Author Response
The authors demonstrated the fabrication of chemically stable Ta2O5 coatings on Ta substrates with micrometer thickness by using micro-arc discharges during electrochemical anodization, able to use for photocatalytic degradation of methyl orange organic compound under UV irradiation. The experiments are well-prepared and the structure of the paper is fine and has the potential to be published in the Micromachines journal. However, there are some issues that will need to be addressed before becoming suitable for publication.
1) Please include in the paper a statistical analysis, i.e. on EDX and organic pollutant decontamination section (Std. deviation or errors bars).
Answer: We have added to the text that the relative errors are less than 5% for EDS data in Table 1.
Table 1 displays the results of the integral EDS analyses of surface coatings (the relative errors are less than 5%) shown in Figure 4a.
Also, we have added the information about the number of tested samples into the manuscript and reproducibility of PA.
Three samples were tested for each processing time, with the mean values shown in Fig. 7a. The PA of samples obtained under the same conditions is highly reproducible (within 3 %).
2) The XRD section must be improved, it looks very poor described.
Answer: Thank you for this comment. We have improved XRD section.
It is widely accepted that Ta2O5 has two main polymorphs, which are usually referred to as high-temperature and low-temperature phases [23]. The transition usually occurs at 1360 oC and this process is reversible, which means that the high-temperature phase cannot be stabilized. As a result, the low-temperature phase is more appealing because it can exist at ambient temperatures. It is usually existing in the form of an orthorhombic or hexagonal crystal structure, with the former one being more stable. The XRD patterns of formed coatings are presented in Figure 5. Orthorhombic Ta2O5 (JCPDS, No. 25-0922) is identified as the main crystalline phase in all of the coatings. This suggests that the rapid solidification of molten Ta2O5 flowing out of the MD channels in the presence of a low temperature electrolyte promotes the formation of orthorhombic Ta2O5. The XRD pattern of coating formed for 15 min also contains a few diffraction lines with low in-tensities, probably connected with tantalum phosphate phases.
3) On the XPS part a calibration information about the system must be included, the position of the adventitious C 1s position for a better understanding for the other elements states.
Answer: We have added calibration information about the system.
Binding energies were corrected relative to the C 1s signal at 285.0 eV.
4) The O 1s peak from high resolution XPS spectrum could contain one more component by fitting it. This is also described by authors on the photoluminescence section where a presumption of oxygen vacancies or presence of P-O binding in the obtained metal-oxide structure is discussed.
Answer: This is a very useful comment. Detailed analysis showed that he high resolution O 1s XPS spectrum can deconvoluted into two components at 530.37 eV and 531.35 eV. We have added in text:
The high resolution O 1s XPS spectrum can be deconvoluted into two components at 530.37 eV and 531.35 eV, indicating two distinct oxide environments: oxygen bonded to phosphorous (530.37 eV) and oxygen bonded to tantalum (531.35 eV) [48].
5) From Figure 4b is not clear the morphology of obtained oxide layer. At list for 3 min and 5 min of anodization, top view (Fig. 4a) reveals a granular topography. It is expected to be distinguished also in the cross-section view. I think that authors used BSE detector in cross-section study, having much less resolution than SE detector. Can the authors clarify this issue?
Answer: Actually, the characterization in Figure 4b was done using the SE detector. BSE detector would probably give better images, but at the moment this detector is inoperable on our device. Polishing during sample preparation most likely destroyed the fine structure in the cross-section view of the samples.
Reviewer 2 Report
Comments to the Author
In this paper, the authors used micro-arc discharges (MDs) technology to acquire Ta2O5 coatings during anodization on a tantalum substrate in a sodium phosphate electrolyte and took it as a photocatalyst to degrade organic pollutants. This work used micro-arc discharges technology to design and prepare photocatalytic coatings. Therefore, the current study is on a topic of relevance and general interest to the readers of the journal. However, there are several issues needing to be addressed before the acceptance of this manuscript in Micromachines:
Specific comments
Comment 1. Page 2, “Plasma-chemical reactions occur in the MD channels as a result of these processes. These reactions raise the pressure inside the MD channels”, could the authors explain why these reactions increase pressure and how they will affect the Ta2O5 coatings?
Comment 2. Page 3, could the authors provide a spectral comparison between the incandescent lamp and the sun, since incandescent lamps are rarely used as solar sources in the field of photocatalysis.
Comment 3. Figure 1, could the authors explain why there is no voltage drop during the first breakdown after stage I.
Comment 4. Figure 4. (a), the figure shows that there are still many holes and defects on the surface after a 15-minute MDs treatment, so could the authors explain why not try a longer MDs time to obtain a smoother surface?
Comment 5. Figure 6. (b) and (c), the authors had better add the splitting peaks information to the high-resolution XPS spectra.
Comment 6. Figure 7. (a), the authors had better add the dark adsorption part in the photocatalytic degradation curve.
Comment 7. Figure 8, could the authors explain why the 15-minute MDs treatment samples have significantly enhanced UV light absorption?
Comment 8. Page 8. “However, during the photocatalysis, oxygen vacancies and defects may serve as capture sites for photoinduced electrons, effectively inhibiting photo-induced electron and hole recombination, which leads to an increase in PA of Ta2O5 coatings.” PL spectra directly show that the recombination of photogenerated carriers was positively correlated with MDs time, which is inconsistent with this statement. Could the authors explain why?

Author Response
In this paper, the authors used micro-arc discharges (MDs) technology to acquire Ta2O5 coatings during anodization on a tantalum substrate in a sodium phosphate electrolyte and took it as a photocatalyst to degrade organic pollutants. This work used micro-arc discharges technology to design and prepare photocatalytic coatings. Therefore, the current study is on a topic of relevance and general interest to the readers of the journal. However, there are several issues needing to be addressed before the acceptance of this manuscript in Micromachines:
Specific comments
Comment 1. Page 2, “Plasma-chemical reactions occur in the MD channels as a result of these processes. These reactions raise the pressure inside the MD channels”, could the authors explain why these reactions increase pressure and how they will affect the Ta2O5 coatings?
Answer: For the sake of clarity, we have removed this sentence.
Comment 2. Page 3, could the authors provide a spectral comparison between the incandescent lamp and the sun, since incandescent lamps are rarely used as solar sources in the field of photocatalysis.
Answer: Thank you for this comment! By mistake we added the description of the old lamp used in our lab. For photocatalytic measurements we used 300 W OSRAM ULTRA-VITALUX UV-A lamp that simulated solar radiation. We corrected this in the text.
Comment 3. Figure 1, could the authors explain why there is no voltage drop during the first breakdown after stage I.
Answer: There is no reason for the voltage drop during the first breakdown after stage I, but the voltage-time slope drops.
Comment 4. Figure 4. (a), the figure shows that there are still many holes and defects on the surface after a 15-minute MDs treatment, so could the authors explain why not try a longer MDs time to obtain a smoother surface?
Answer: Thank you for this comment. We attempted a longer PEO time, but the MDs gradually vanished after 15 minutes, rendering the process ineffective.
Comment 5. Figure 6. (b) and (c), the authors had better add the splitting peaks information to the high-resolution XPS spectra.
Answer: We have added the splitting peaks information to the high-resolution XPS spectra.
Comment 6. Figure 7. (a), the authors had better add the dark adsorption part in the photocatalytic degradation curve.
Answer: We added the dark adsorption part in the photocatalytic degradation curves.
Comment 7. Figure 8, could the authors explain why the 15-minute MDs treatment samples have significantly enhanced UV light absorption?
Answer: Improved the UV light absorption capacity of Ta2O5 coating formed for 15 min is a results of the highest coating thickness. This statement has been added to the text.
Comment 8. Page 8. “However, during the photocatalysis, oxygen vacancies and defects may serve as capture sites for photoinduced electrons, effectively inhibiting photo-induced electron and hole recombination, which leads to an increase in PA of Ta2O5 coatings.” PL spectra directly show that the recombination of photogenerated carriers was positively correlated with MDs time, which is inconsistent with this statement. Could the authors explain why?
Answer: Thank you for this comment. These are two processes that occur simultaneously. For this reason, the stronger the excitonic PL intensity, the higher the photocatalytic activity. Furthermore, oxygen vacancies can stimulate O2 adsorption, and photo-induced electrons bound by oxygen vacancies interact strongly with adsorbed O2 [56]. This suggests that oxygen vacancies can help adsorbed O2 capture photo-induced electrons while also producing ·O2 radical groups. The radical groups are active in promoting the oxidation of organic substances. Thus, oxygen vacancies and defects may favor photocatalytic reactions, and the stronger the excitonic PL emission, the greater the oxygen vacancy or defect content, and the higher the photocatalytic activity.
We have added this paragraph in text.
Round 2
Reviewer 1 Report
The manuscript looks better after review and can be considered for publication in present form. My suggestion to the authors, for the future work to clarify with the SEM analysis in the cross-section view. To better discover the morphology after the polishing, a chemical etching or sonication is required.